# Coagulopathy in the Setting of Mild Traumatic Brain Injury: Truths and Consequences

**DOI:** 10.3390/brainsci7070092

**Published:** 2017-07-22

**Authors:** Joseph P. Herbert, Andrew R. Guillotte, Richard D. Hammer, N. Scott Litofsky

**Affiliations:** 1Division of Neurological Surgery, University of Missouri School of Medicine, Columbia, MO 65212, USA; herbertj@health.missouri.edu (J.P.H.); arg553@health.missouri.edu (A.R.G.); 2Department of Pathology and Anatomical Sciences, University of Missouri School of Medicine, Columbia, MO 65212, USA; hammerrd@health.missouri.edu

**Keywords:** traumatic brain injury, concussion, coagulopathy, thromboelastography, platelet dysfunction

## Abstract

Mild traumatic brain injury (mTBI) is a common, although poorly-defined clinical entity. Despite its initially mild presentation, patients with mTBI can rapidly deteriorate, often due to significant expansion of intracranial hemorrhage. TBI-associated coagulopathy is the topic of significant clinical and basic science research. Unlike trauma-induced coagulopathy (TIC), TBI-associated coagulopathy does not generally follow widespread injury or global hypoperfusion, suggesting a distinct pathogenesis. Although the fundamental mechanisms of TBI-associated coagulopathy are far from clearly elucidated, several candidate molecules (tissue plasminogen activator (tPA), urokinase plasminogen activator (uPA), tissue factor (TF), and brain-derived microparticles (BDMP)) have been proposed which might explain how even minor brain injury can induce local and systemic coagulopathy. Here, we review the incidence, proposed mechanisms, and common clinical tests relevant to mTBI-associated coagulopathy and briefly summarize our own institutional experience in addition to identifying areas for further research.

## 1. Introduction

Coagulopathy occurs frequently after traumatic brain injury (TBI). Coagulopathy can affect outcome and contribute to sequela after the initial injury. Most study of coagulopathy has been in patients with severe TBI. However, mild TBI (mTBI) accounts for about 70%–90% of TBI patients, depending on one’s definition of mTBI [1]. Understanding of role of coagulopathy in this significant subset of patients is therefore essential. This review focuses on the incidence and consequences of coagulopathy in patients with mTBI.

## 2. Definition of Mild Traumatic Brain Injury

mTBI can be defined in a number of different manners. Classical categorization of mTBI has included patients with Glasgow Coma Scale (GCS) of 13–15. Others have broadly characterized mTBI as the sequelae of neurotrauma leading to mild, transient alterations of consciousness without structural abnormality on neuroimaging [2,3,4,5]. Some have even gone so far as to question its existence as a distinct clinical entity [6]. Concussion has been included as one type of mTBI or synonymous with mTBI, depending on the source [2,3,7,8]. The Brain Trauma Foundation, which has published guidelines for the management of severe TBI (defined as GCS 3–8), indicates that mTBI is synonymous with concussion, stating that prevalent and consistent indicators of concussion include “(1) observed and documented disorientation or confusion immediately after the event, (2) impaired balance within 1 day after injury, (3) slower reaction time within 2 days after injury, and/or (4) impaired verbal learning and memory within 2 days after injury” [9]. Recent research into characterization and treatment of mTBI has been largely based on experiences in contact sports and the military [3,7,8,9,10,11]. From 2000–2016, mTBI accounted for 82.4% of all TBI in the military [12]. For the purposes of the discussion that follows, we will define mTBI as patients with post-resuscitation GCS of 13–15, regardless of head computed tomography (CT) abnormalities or not.

## 3. Coagulopathy in Traumatic Brain Injury

The characterization, underlying mechanism(s), and treatment of coagulopathy in trauma patients, or trauma-induced coagulopathy (TIC), remains an elusive target despite intensive research from both civilian and military centers [13,14,15,16,17,18,19,20,21,22,23]. Despite countless original articles and several large-scale trials, there is no clear understanding of the molecular underpinnings of coagulopathy in trauma patients, nor is there yet a consensus on how best to treat such patients. What can at times seem a bewildering jungle of data arguing for and against various putative causative agents has been succinctly reviewed in articles by Chang et al. [13] and Walsh et al. [15], with Chang et al. concluding, “… as investigation of TIC proceeds, its complexity seems to deepen.” The development of TIC appears to involve a complex interplay between various procoagulant, anticoagulant and fibrinolytic pathways eventually leading to platelet dysfunction and exhaustion of clotting factors, the exact nature and timing of which is both beyond the scope of this review and the subject of vigorous debate within the trauma community. 

Coagulopathy related to TBI was first reported nearly 60 years ago [24] and has since been demonstrated in both blunt and penetrating isolated TBI; coagulopathy is heavily represented in the trauma literature [25,26,27,28,29,30,31,32,33,34]. Coagulopathy is more common with increasing severity of injury [33] and is associated with a worse prognosis [35,36,37] in TBI patients. Despite this increased research interest, the underlying mechanism of TBI-induced coagulopathy and its relationship to TIC is unclear. Several mechanisms have been proposed, as discussed in the next section.

## 4. Proposed Mechanisms

Isolated TBI generally does not involve systemic shock unlike that which is often seen in TIC. Therefore, it is unlikely that TBI-induced coagulopathy depends heavily on activated protein C (APC), a commonly-cited potential mechanism for TIC (particularly for early TIC) which is strongly correlated to hypoperfusion in trauma victims [17,20,21,29]. The procoagulant tissue factor (TF) and anticoagulant tissue plasminogen activator (tPA) have long been thought to play a central role in both TIC and TBI-induced coagulopathy. Brain tissue itself is rich in TF [19], which is a transmembrane receptor for factor VII/VIIa and plays a central role in the activation of the clotting cascade. TF is thought to act to induce a procoagulant state immediately following injury with consequent delayed consumption of clotting factors leading to a hypocoagulable state. This mechanism may explain, in part, the delayed hemorrhage often seen in cerebral contusions. However, Hijazi et al. [38] demonstrate in a mouse model that tPA and urokinase plasminogen activator (uPA), not TF, is central in mediating coagulopathy following TBI. In their study, tPA and uPA knockout mice are shown to have decreased progressive intracerebral hemorrhage (ICH) compared with the wild-type (WT). Additionally, mice premedicated with warfarin also show decreased progressive ICH when treated with an inhibitor of tPA. Importantly, systemic coagulopathy is seen in tPA knockout mice, but is not seen in uPA knockout mice. In a separate experiment, the authors note that uPA levels in CSF peak in a delayed fashion approximately 9 h after injury and remain elevated up to 36 h post-injury; tPA, on the other hand, peaks almost immediately after injury and tapers off quite rapidly. This suggests that uPA may play a central role in mediating post-TBI systemic coagulopathy. A subsequent study by Chapman et al. [39] demonstrates that trauma patients with TEG-confirmed fibrinolysis have significantly elevated levels of tPA, but similar levels of plasminogen activation inhibitor (PAI-1, a target for aPC) compared with controls, suggestive of the central role of tPA rather than aPC in TIC. 

The role of TF in post-TBI coagulopathy should not be entirely discounted, however. Microparticles (MP) are circulating phospholipid vesicles less than 1 μm in diameter which are derived from platelets, endothelial cells, as well as other hematopoietic cells [40] and, more recently recognized, neuronal cells [41], which display a variety of surface antigens, including procoagulant MPs displaying TF [42]. MPs have been shown to be elevated following trauma [14,43,44] as well as isolated TBI [45,46,47,48]. In a small pilot study, Morel et al. [46] show that procoagulant platelet- and endothelial-derived MPs are elevated in both the plasma and CSF of patients with severe TBI up to 10 days post-injury compared with healthy controls. In a revealing 2015 study, Tian et al. [47] look at the role of brain-derived MPs (BDMPs), including both neuron- and glial cell-derived MPs, in post-TBI coagulopathy. They demonstrate that blood levels of BDMPs peaked at 3 h post-TBI, with continued elevated levels of both neuron- and glial-derived MPs evident even 6 h after injury. The study mice have systemic hypercoagulopathy within 3 h of injury, which reverses with removal of BDMPs. Interestingly, healthy mice injected with purified BDMPs at low levels develop systemic hypercoagulability while those injected with higher levels of BDMPs transition to a consumptive hypocoagulability in a dose-dependent fashion. BDMPs display both phosphatidylserine (PS) and TF on their surfaces. Surprisingly, although BDMPs activate human platelets in vitro, they do directly induce platelet aggregation. The authors hypothesize that the failure to induce platelet aggregation is due to steric hindrance of the bound BDMPs preventing direct platelet-to-platelet interaction, but clearly this is a question in need of further research. BDMPs suggest a novel mechanism by which mild TBI could induce coagulopathy; namely, even a small, localized injury to tissue rich in anticoagulant microparticles could induce widespread coagulopathy, thus eliminating the systemic hypoperfusion previously thought to be a hallmark of TIC and offering a potential therapeutic target. 

## 5. Clinical Tests for Coagulopathy

Clinical tests for coagulopathy most commonly include assessments of the coagulation cascade (prothrombin time, the related international normalized ratio, and partial thromboplastin time) and platelet function (thromboelastography). We briefly review these clinical laboratory tests and their limitations. 

Prothrombin time (PT), international normalized ratio (INR, which is derived from measured PT and controls for variations in tissue factors used in various manufacturers’ reagents) and partial thromboplastin time (PTT) are tests of the coagulation cascade and were initially designed to determine clotting factor deficiencies [9]. While commonly used in the clinical setting, particularly for monitoring of pharmacologic anticoagulation, they only reflect time before initial thrombin generation and are poor indicators of acute, acquired coagulopathy [9,49,50] and furthermore do not detect hypercoagulability [23]. For these reasons, PT and PTT are not generally used as indicators of TIC in modern trauma research [50]. 

Thromboelastography (TEG), or Rotational thromboelastography (ROTEM), a nearly identical assay which has only minor differences in mechanics, utilizes the same principle of clot formation and renders similar data. It is sometimes studied in tandem with TEG (see Ref. [50] for example).), was first developed in 1948 and has since been widely adopted in clinical practice, particularly in trauma. While a detailed description of the mechanics of TEG is not pertinent to this review, the fundamental principle is fairly straightforward: clotting is the end result of the coagulation cascade, and measurement of various clot properties can determine presence or absence of coagulopathy. Clot strength, fibrinolysis, and degree of platelet inhibition are measured by mechanically agitating a sample of blood [51]. A further refinement is the TEG PlateletMapping assay (Haemonetics, Braintree, MA) which includes analysis of platelet dysfunction due to inhibition via the arachidonic acid (AA) pathway (as in the case of aspirin) or the adenosine diphosphate (ADP) pathway (as in clopidogrel) [52].

## 6. Incidence

The estimated incidence of TBI-induced coagulopathy varies widely in the literature. A 2008 systematic review by Harhangi et al. [32] is perhaps the most commonly cited authority on the subject. They reviewed 82 articles regarding TBI-associated coagulopathy over a 40-year period including 34 studies reporting frequency and arrived at an overall incidence of 32.7%, with a wide variation noted between studies. These discrepancies are likely attributable to the wide variety of definitions of both TBI and coagulopathy used over the years. In their review, approximately 60% patients with severe TBI [22,53] were found to be coagulopathic, compared with less than 1% of patients with mild TBI [54]. For the purposes of this review, however, the 1% figure is from a large retrospective study which defined mild TBI as GCS 13–15 which can include patients with abnormalities on CT head. Under the current DoD criteria, these patients would be classed as “moderate” TBI [12]. Such classification discrepancies further complicate any endeavor to determine the actual prevalence of TBI-associated coagulopathy.

We collected TEG with platelet mapping (PM) data at the University of Missouri Hospital and Clinics as part of an ongoing quality improvement study evaluating TEG abnormalities in TBI patients [55]. Adult patients with evidence of traumatic injury detected on computed tomography (CT) after blunt head trauma were included in the study. Exclusions from the study were age younger than 18 years and penetrating head trauma. Ninety-seven patients who met the criteria for inclusion in the study were not taking antiplatelet or anticoagulant medications at the time of injury. Eighty patients in the study had TEG/PM assays performed at admission. Fifty-four (67.5%) of those patients were classified as mild TBI by GCS ranging from 13 to 15. Platelet dysfunction was defined as ADP pathway inhibition greater than 60% by TEG/PM. ADP inhibition greater than 60% has been shown to result in increased bleeding in cardiac surgery patients [56]. Seventeen (31.5%) of the mTBI patients had ADP inhibition greater than 60%. The results are summarized in Table 1 divided into groups for the total study sample, isolated mild TBI, and mild TBI that occurred in conjunction with trauma to other organ systems (non-isolated TBI). Platelet ADP inhibition appeared to be much more common compared to the healthy controls studied previously at the University of Missouri Hospital and Clinics as part of a study conducted by Bartels et al. [57] (Table 2). The control group had mean ADP pathway inhibition of 7.4% (standard deviation = 8.3). The study sample of isolated mTBI patients had a mean ADP inhibition of 42.3% (standard deviation = 27.9). The difference between the study sample and the controls was statistically significant (*p* = 1.194 × 10^−6^, two-tailed *t*-test). 

Thus, even mild TBI can be associated with platelet dysfunction. Questions that remain to be answered include whether this level of platelet dysfunction is a risk factor for poor outcomes in TBI patients. If platelet dysfunction does result in poor outcomes, how should the patients be managed?

We also examined traditional tests for coagulopathy, including PT, PTT, and platelet count in 142 of the patients included in our TEG/TBI project at the University of Missouri Hospital and Clinics (Table 3). Again, all patients included in this study had CT abnormalities attributed to blunt head trauma. Patients with incomplete laboratory results were excluded from analysis. Patients were considered to have coagulopathy if their PT was greater than 15.9 s, PTT was greater than 37.2 s, or the platelet count was less than 100 × 10^3^ /microliter. Mild TBI was defined as GCS 13–15, moderate TBI was GCS 8–12, and severe TBI was GCS 3–7. The limits for determining coagulopathy for PT and PTT were collected and determined by the University of Missouri Hospital laboratory [58]. The data was used to set the reference ranges for the PT and PTT laboratory tests for patient care. Figure 1 shows the median and standard deviations of the test results. 

We additionally examined the cohort of patients with isolated TBI who were not taking anti-platelet or anticoagulant medications (Table 4). Control data was derived by the University of Missouri Hospital laboratory [58] to set the reference ranges for the PT and PTT laboratory tests for patient care. Although the sample size was small for moderate and severe TBI groups, this data did not show a higher incidence of coagulopathy in isolated TBI patients compared to controls using the traditional coagulation measures. Other injuries and medications appeared to contribute to much of the coagulopathy seen in mild TBI determined by traditional testing. 

## 7. Consequence of Coagulopathy in Mild TBI

Although the vast majority of patients with mild TBI have uneventful clinical courses and improve back to their baseline with time, a small subset goes on to develop a feared complication known as delayed neurological deterioration (DND), known colloquially as the “talk-and-die” phenomenon [59]. The exact definition of this phenomenon varies in the literature, but generally refers to patients who present with fairly minor symptoms (GCS ≥13 or a verbal GCS of ≥3, for example) who subsequently have a significant neurological decline in a delayed fashion. It is generally thought that the demise of such patients is mainly due to progression of their initial minor-appearing brain injuries [60]. In a large retrospective study of 7443 TBI patients in San Diego County who had a verbal GCS score of ≥3 at some point in their preadmission course, Davis et al. [61] report an overall mortality of 6.1%, with nearly one-third occurring in the first hospital day. They state that risk factors for early death include severity of extracranial injuries and, importantly, use of anticoagulants.

A more recent single-center retrospective study by Choudhry et al. [62] examined patients who presented with mild head injury defined as GCS ≥13 with acute hemorrhage on CT head. Of the 757 patients who met inclusion criteria, 4.1% experienced DND, mostly as a result of progression of intracranial hematoma. Logistic regression analysis found that coagulopathy on admission (defined by elevated INR, PT, PTT or platelets <100,000) was an independent risk factor for DND (*p* < 0.02). Twenty-three percent of patients with DND died, all within 12 h of admission, underscoring the importance of anticipation and early recognition in the acute management of these patients.

## 8. Future Directions

Despite its recognition even in ancient times, the management of traumatic brain injury is in many ways still in its infancy. Widespread disagreement and controversy exists over even minor classification criteria and treatment protocols. Over the last several decades, however, there have been paradigm-shifting advances in both the basic science and clinical practice of TBI management. There are several areas in particular where more research is needed to guide clinical practice.

The fundamental molecular biology underlying TBI-induced is far from fully elucidated. The research into the roles of tPA and BDMPs which are briefly reviewed above offer potential therapeutic targets for prevention and treatment of TBI-induced coagulopathy in addition to deepening our understanding of the underlying mechanisms of platelet activation. The animal studies that are briefly reviewed in this article could form the basis of prospectively collected data from human TBI patients.

Prophylactic anticoagulation using subcutaneous unfractionated heparin or low-molecular weight heparin has become routine in American hospitals for prevention of deep venous thrombosis (DVT) and pulmonary embolism (PE). Indeed, pulmonary embolism was found to be a small but significant contributor to late mortality in TBI patients in the San Diego study [61]. Optimal timing of such prophylaxis is especially controversial in TBI patients, however [63]. No randomized controlled trial designed to answer what type and duration of prophylactic anticoagulation is optimal in this patient population has been performed. 

What is the appropriate treatment of TBI-associated coagulopathy, and what should the treatment parameters be? The most recent Brain Trauma Foundation guidelines for management of severe TBI do not make any comment regarding therapeutic strategies for TBI-associated coagulopathy [64]. Some retrospective studies advocate the use of recombinant factor VIIa (rFVIIa) [65] or prothrombin complex concentrate (PCC) [66]. Such therapies have a sound physiologic basis, but there are currently no published data examining interventions for TBI-associated coagulopathy in a randomized, controlled fashion.

## 9. Conclusions

TBI-induced coagulopathy remains a significant and probably under-recognized contributor to early mortality in patients with head injury. The current body of research demonstrates that even mild TBI can cause clinically significant coagulopathy, particularly with regard to platelet dysfunction, while also highlighting the considerable gaps in our basic knowledge of its pathogenesis and molecular biology. 

## Figures and Tables

**Figure 1 brainsci-07-00092-f001:**
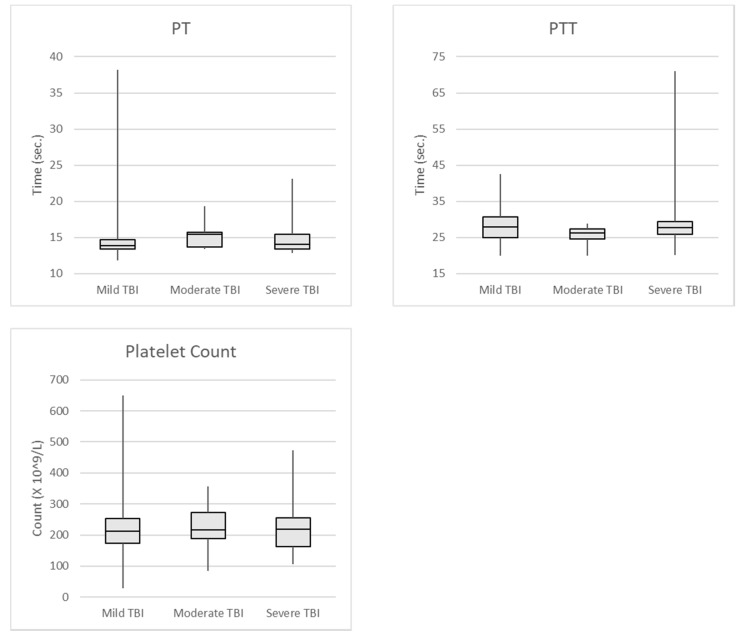
PT, PTT, and platelet count in all TBI Patients.

**Table 1 brainsci-07-00092-t001:** Platelet Inhibition in mild traumatic brain injury (TBI) patients with computed tomography (CT) abnormalities.

	All TBI Patients *N* (%)	Isolated TBI *N* (%)	Non-Isolated TBI *N* (%)
TBI patients included in analysis	97	43	54
TBI patients with TEG/PM assays	80	36	44
Patients with mild TBI	54 (67.5%)	29 (80.6%)	25 (56.8%)
mTBI with adenosine diphosphate (ADP) inhibition ≥60%	17 (31.5%)	7 (24.1%)	10 (40.0%)

Patients taking antiplatelet or anticoagulant medications were excluded from analysis.

**Table 2 brainsci-07-00092-t002:** Platelet ADP inhibition in isolated mTBI patients with CT abnormalities compared to controls.

	*N*	Mean	Standard Deviation	95% CI
Control [57]	8	7.4	8.3	±6.9
Isolated mTBI ^1^	29	42.3	27.9	±10.6

^1^ Statistically significant difference (*p* = 1.194 × 10^−6^, two-tailed *t*-test). Patients taking antiplatelet or anticoagulant medications were excluded from analysis.

**Table 3 brainsci-07-00092-t003:** Coagulopathy by prothrombin time (PT), partial thromboplastin time (PTT), and international normalized ratio (INR) in all TBI patients

	*N*	Prolonged PT	Prolonged PTT	Thrombocytopenic	Coagulopathic by at Least One Parameter
Mild TBI	102	16 (15.7%)	4 (3.9%)	20 (19.6%)	33 (32.4%)
Moderate TBI	11	6 (54.5%)	0	2 (18.2%)	6 (54.5%)
Severe TBI	29	11 (37.9%)	2 (6.9%)	0	14 (48.3%)

**Table 4 brainsci-07-00092-t004:** Coagulopathy by PT, PTT, and platelet count in isolated TBI patients with CT abnormalities.

	*N*	Prolonged PT	Prolonged PTT	Thrombocytopenic
Control	40	2 (5%)	1 (2.5%)	-
Mild TBI	32	0	0	2 (6.3%)
Moderate TBI	3	1 (33.3%)	0	1 (33.3%)
Severe TBI	4	0	0	0

Patients taking antiplatelet or anticoagulant medications were excluded from analysis.

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
