# Peer review of "Coagulopathy in the Setting of Mild Traumatic Brain Injury: Truths and Consequences"

_brainsci, 2017, doi:10.3390/brainsci7070092_

Round 1
Reviewer 1 Report
1. Authors evaluated TEG abnormalities in TBI patients of their institute. The patients with oral antiplatelet agents and anticoagulants should be excluded before evaluation.
The rate of coagulopathic mTBI patients included in this criteria should be shown in this paper.
2. We’d like to know the information about these TBI patients are isolated TBI or not. Because coagulopathy is induced with different mechanisms in multiple trauma patients compared with isolated TBI.
3. Table 3 and Figure 1 are the same contents, so one of 1 are fine.
Author Response
REVIEWER #1
1. Authors evaluated TEG abnormalities in TBI patients of their institute. The patients with oral antiplatelet agents and anticoagulants should be excluded before evaluation. The rate of coagulopathic mTBI patients included in this criteria should be shown in this paper.
We agree that anti-platelet and anticoagulant medications confounds the issues. Therefore, report the data excluding those patients in a new Table 1 on page 4, line 173, and a new Table 2 on page 5, line 177. However, we also maintain reporting the data without excluding those patients because they do represent a significant clinical population with mild traumatic brain injury
2. We’d like to know the information about these TBI patients are isolated TBI or not. Because coagulopathy is induced with different mechanisms in multiple trauma patients compared with isolated TBI.
Table 1 lists ADP inhibition in both patients with isolated TBI and those with multi-trauma. Table 2 shows ADP inhibition in isolated mTBI and non-injured patients.
3. Table 3 and Figure 1 are the same contents, so one of 1 are fine.
Table 3 has been removed. Figure 1 was maintained
Reviewer 2 Report
Although historically there was some confusion regarding the definition of mTBI recent consensus defines the entity as presentation with a GCS of 13-15 with no abnormalities noted on CT scan of the brain. ( Essentially concussion) This paper does not clearly define the patient population for the reader. How many of their patients had abnormalities on their admission CT scan , or presented with a GCS<13? These are critical issues. Additionally, these results are not statistically compared against the author's normal's. Would this not add significantly to the veracity of the conclusions indicating mTBI is a causative factor in this patient population. Finally, did these patients have any concomitant injuries which may have contributed to the elevated coagulopathy?
Author Response
REVIEWER #2
Although historically there was some confusion regarding the definition of mTBI recent consensus defines the entity as presentation with a GCS of 13-15 with no abnormalities noted on CT scan of the brain. ( Essentially concussion) This paper does not clearly define the patient population for the reader. How many of their patients had abnormalities on their admission CT scan , or presented with a GCS<13? These are critical issues. Additionally, these results are not statistically compared against the author's normal's. Would this not add significantly to the veracity of the conclusions indicating mTBI is a causative factor in this patient population. Finally, did these patients have any concomitant injuries which may have contributed to the elevated coagulopathy?
We clearly state that for the purposes of our discussion and data included that we define mild traumatic brain injury as GCS 13-15, regardless of CT abnormalities or not. All patients included in our TEG analysis data had CT abnormalities (Page 4, line 157). We also show data for patients with moderate and severe TBI (GCS < 13) in Table 3 (Page 5, line 197) and Table 4 (Page 6, line 222). We report normals at University of Missouri for TEG in Table 2 (Page 5, line 177) and Table 4 (Page 6, line 222). Table 1 (page 4, line 173) lists ADP inhibition in both patients with isolated TBI and those with multi-trauma. Table 2 shows ADP inhibition in isolated mTBI and non-injured patients.
Round 2
Reviewer 1 Report
Authors are correcting sufficiently.